# Knowledge Beacons: Web services for data harvesting of distributed biomedical knowledge

**Lance M. Hannestad[1,2], Vlado Dančík[3], Meera Godden[1,2], Imelda W. Suen[1,4], Kenneth C. Huellas-Bruskiewicz[1,5], Benjamin M. Good[6], Christopher J. Mungall[6‡], Richard M. Bruskiewich** **[1‡]\***

1 STAR Informatics / Delphinai Corporation, Sooke, BC, Canada, 2 Department of Computing Science, Simon Fraser University, Burnaby, BC, Canada, 3 Chemical Biology and Therapeutics Science Program, The Broad Institute, Cambridge, MA, United States of America, 4 School of Computing Science, University of British Columbia, Vancouver, BC, Canada, 5 School of Interactive Arts and Technology, Simon Fraser University, Burnaby, BC, Canada, 6 Lawrence Berkeley National Laboratory, Berkeley, CA, United States of America

‡ These authors are joint senior authors on this work.
* richard.bruskiewich@delphinai.com

**Data Availability Statement:** This project has not generated any novel data of its own but provides access to external public resources using software

## Abstract

The continually expanding distributed global compendium of biomedical knowledge is diffuse, heterogeneous and huge, posing a serious challenge for biomedical researchers in knowledge harvesting: accessing, compiling, integrating and interpreting data, information and knowledge. In order to accelerate research towards effective medical treatments and optimizing health, it is critical that efficient and automated tools for identifying key research concepts and their experimentally discovered interrelationships are developed. As an activity within the feasibility phase of a project called "Translator" (**https://ncats.nih.gov/translator**) funded by the National Center for Advancing Translational Sciences (NCATS) to develop a biomedical science knowledge management platform, we designed a Representational State Transfer (REST) web services Application Programming Interface (API) specification, which we call a Knowledge Beacon. Knowledge Beacons provide a standardized basic API for the discovery of concepts, their relationships and associated supporting evidence from distributed online repositories of biomedical knowledge. This specification also enforces the annotation of knowledge concepts and statements to the NCATS endorsed the Biolink Model data model and semantic encoding standards (**https://biolink.github.io/biolink-model/**). Implementation of this API on top of diverse knowledge sources potentially enables their uniform integration behind client software which will facilitate research access and integration of biomedical knowledge.

## Availability

The API and associated software is open source and currently available for access at **https://github.com/NCATS-Tangerine/translator-knowledge-beacon**.

published in Github software repositories whose links are given within the manuscript.

**Funding:** The Knowledge Beacon API was primarily developed by the STAR Informatics / Delphinai Corporation team, in collaboration with research collaborators at the Scripps Institute, the Lawrence Berkeley Laboratories and the Broad Institute, as a subcontracted activity of the Biomedical Data Translator Consortium, through funding provided by the National Center for Advancing Translational Sciences, National Institutes of Health (Co-author VD supported by NCATS grants 1OT2TR002584 and OT3TR002025; all other authors received support under NCATS grant 1OT3TR002019) NCATS, the funding organization, provided support in the form of project funding for salaries for every author on this paper, but did not have any additional role in the study design, data collection and analysis, decision to publish, or preparation of the manuscript. The specific roles of these authors are articulated in the 'author contributions' section. The commercial role of authors – both regular staff and cooperative education students - working under the auspices of STAR Informatics / Delphinai Corporation - a small private Canadian federally incorporated scientific informatics business located in British Columbia, Canada - was simply to deliver software and data curation outputs on a staff time billable consultancy basis, in accordance with Biomedical Data Translator project objectives, outputs which have no other further significant marketable commercial benefit to the firm. Any opinions expressed in this document are those of the authors and may not necessarily reflect the broader views of The Biomedical Data Translator Consortium, the National Institutes of Health, other individual Translator team members, nor affiliated organizations and institutions.

**Competing interests:** The commercial role of RMB within STAR Informatics / Delphinai Corporation is one of Founder CEO and Principal Scientist of the firm. All other authors working for the firm conducted the work under regular full-time or temporary employment contract basis, under primary supervision by RMB. The Biomedical Data Translator project outputs for this paper, open sourced and access in nature, as delivered by the consultancy subcontract, do not represent any other marketable commercial benefit to the firm in terms of patents, products in development, or marketed products. This does not alter our adherence to PLOS ONE policies on sharing data and materials.

## Introduction

A serious challenge to impactful biomedical research is the one that biomedical researchers encounter when identifying and accessing pertinent information: the diffuse and voluminous nature of such data and knowledge. The large, rapidly growing compendium of published scientific literature is characterized by diverse data encoding standards; numerous, distinct, heterogeneous, large and often siloed public research data repositories; relatively inaccessible health records; numerous clinical trial and adverse event reports, all spread across disease communities and biomedical disciplines. The current distributed nature of this knowledge and associated (meta-)data silos impedes the discovery of related concepts and the relationships between them, an activity one might call "Knowledge Harvesting". Many efforts to overcome this challenge focus on data management principles to make such resources "**F**indable, **I**nteroperable, **A**ccessible and **R**eusable" (FAIR) [1, 2].

Web access to bioinformatics data spans many generations of web service standards tagged with many acronyms, e.g. CORBA [3], SOAP/BioMOBY [4] and SADI [5], the latter an exemplar of the more general paradigm of "Linked Open Data" using OWL/RDF and SPARQL technology, including Linked Open Fragments [6].

A popular web service standard currently in use is the Swagger 2.0 or OpenAPI 3.0 specified REST API (https://github.com/OAI). Many extant online biomedical data sources currently provide such REST API implementations for accessing their data. API registries exist to index such APIs to facilitate access (for example, the Smart API Registry; https://smart-api.info/) and generalized tools are available to explore the space of such web services (notably, the Biothings API and Explorer; https://biothings.io/). However, the heterogeneity of such APIs can be a barrier to efficient biomedical knowledge integration.

Here we present a REST-based web services specification called the Knowledge Beacon API (Beacon API) that enables discovery of, and navigation through, biomedical concepts, relationships and associated evidence. This work arises out of an earlier effort to develop a web application called "*Knowledge.Bio*" [7] to provide enhanced navigation through the knowledge base of PubMed cited concepts and relationships, captured by text mining in the Semantic Medline Database [8]. The knowledge harvesting activity underlying *Knowledge.Bio* is here elaborated into a distributed web service network across diverse knowledge sources hosted within the NCATS Biomedical Data Translator Consortium, a publicly funded project supporting the FAIR integration of distributed biomedical research data and knowledge to accelerate the development of new disease treatments and reduce the barriers between basic research and clinical advances [9]. The outcome of this work was an iteratively refined web service specification implemented in an initial set of Beacons, with validation tools and client applications.

## Methods

The Knowledge Beacon API is a Swagger 2.0 specification that defines a set of endpoint paths embodying operations for accessing knowledge sources and discovering shared semantics for concepts and their relationships (Fig 1; see also https://github.com/NCATS-Tangerine/translator-knowledge-beacon).

A Knowledge Beacon (hereafter abbreviated "Beacon") initiates knowledge discovery by simple search using a concept endpoint either with a *keywords* parameter (/**concepts?keywords** =) or one with a Compact Uniform Resource Identifier (CURIE; https://en.wikipedia.org/wiki/CURIE) of the concept (/**concepts/{conceptId}**). In both cases, one or more specific concepts with associated core details are retrieved.

Once identified, the canonical CURIE identifier of a chosen concept selected from the retrieved list is used as an input parameter to access a list of statements about the concept, documented as

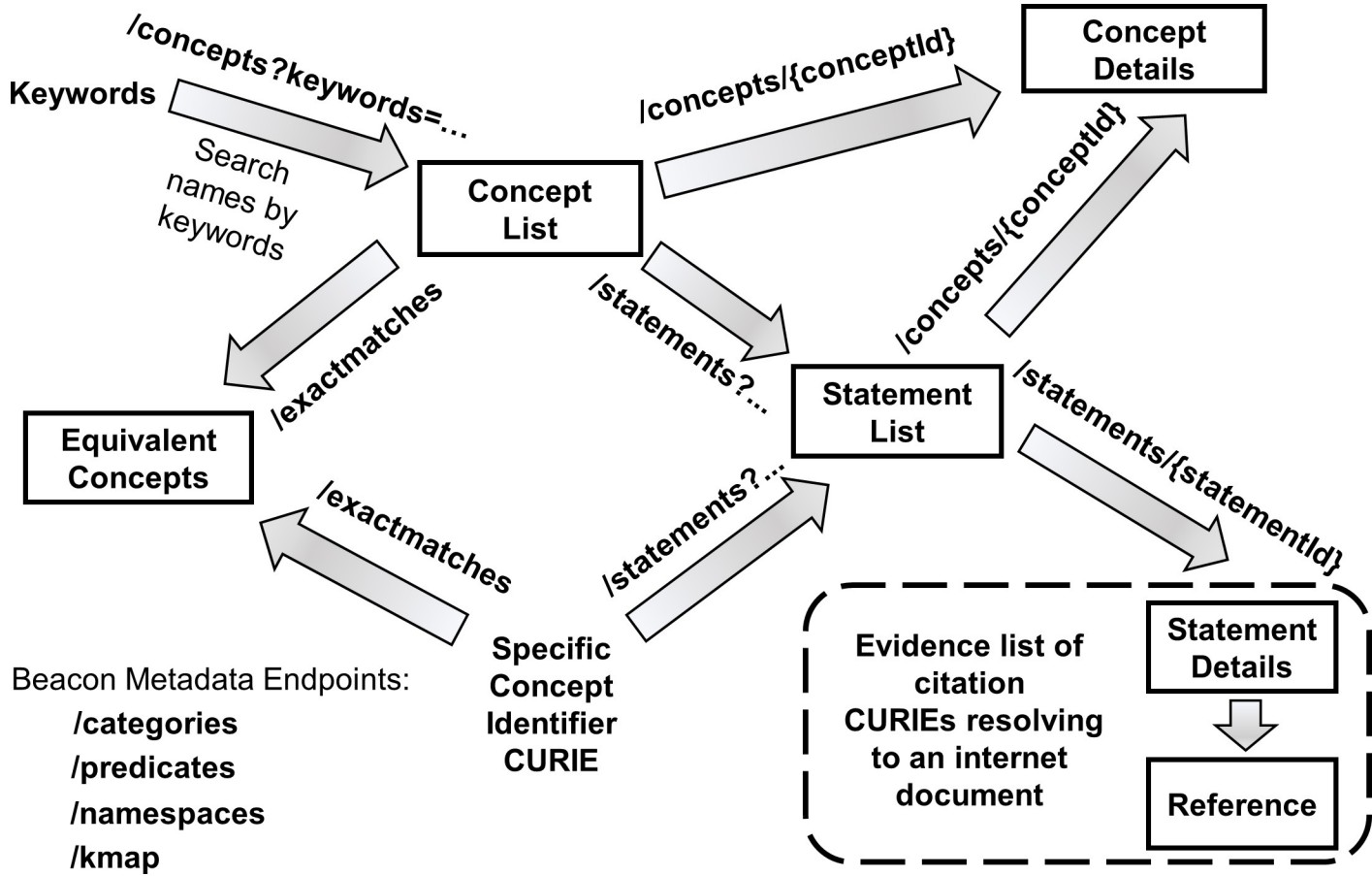

**Fig 1. Knowledge Beacon API.** General step-by-step flowchart illustrating the sequential invocation of Beacon web service endpoints, with data flows as indicated. Also enumerated at the bottom lefthand corner of the diagram is the set of metadata endpoints that report semantic terms and namespaces used by the Beacon in the annotation of results.

subject/predicate/object assertions (**/statements?s =**... where **s** is a **s**ubject canonical concept CURIE). Additional documentation, including supporting citations, associated with returned statements may be examined by calling the statement's endpoint again with the statement identifier of one of the entries returned from the initial call (i.e. **/statements/{statementId}**).

The data model, concept data type ("*category*") and relationship predicate ("*edge_label*", "*relation*") terms in results returned by a Beacon are compliant with an emerging public Biomedical Data Translator Consortium semantic standard and data model, the Biolink Model (**https://biolink.github.io/biolink-model/**). To assist client data parsing and interpretation, a Beacon supports several additional endpoints that return metadata summaries of Biolink Model terms specifically employed by the Beacon to annotate concepts and statements which are returned: concept type "categories" (**/categories**), identifier name spaces (**/namespaces**), relationship "predicates" (**/predicates**) plus a "knowledge map" of available subject-predicate-object triplet statement combinations (**/kmap**).

## Results

### Beacon implementations

A stable set of publicly accessible Beacons are implemented and currently hosted stably online (as of January 2021) by the NCATS Biomedical Translator Consortium, as enumerated in

**Table 1. Biomedical translator consortium deployed beacons.**

| Subdomain[a] | Beacon Description | Wrapped Knowledge Source |
|---|---|---|
| semmeddb | Semantic Medline Database [8] | https://skr3.nlm.nih.gov/SemMedDB/ |
| biolink | Monarch Database Biolink API [10] | https://api.monarchinitiative.org/api/ |
| hmdb | Human Metabolome Database [11] | http://www.hmdb.ca/ |
| rhea | Rhea Annotated Biochemical Reactions database [12] | https://www.rhea-db.org/ |
| smpdb | Small Molecular Pathway Database [13] | http://smpdb.ca/ |
| ndex | NDex Bio Graph Archive [14] | http://www.ndexbio.org |

[a] The basepath of each Beacon has the form **https://kba.ncats.io/beacon/<Subdomain>**, where the <**Subdomain**> is as listed in column 1 of the table.

Table 1. The Java and Python software implementations of these Beacons are available in repositories of the NCATS-Tangerine (https://github.com/NCATS-Tangerine) GitHub organization. One implementation is a generic accessor of Biolink Model compliant knowledge graph databases stored in Neo4j (https://github.com/NCATS-Tangerine/tkg-beacon). These Beacon implementations may be tested using an available validator application (https://github.com/NCATS-Tangerine/beacon-validator). A Python command line Beacon client is available (https://github.com/NCATS-Tangerine/tkbeacon-python-client). A Knowledge Beacon Aggregator (https://github.com/NCATS-Tangerine/beacon-aggregator-client) was also designed to manage a registered pool of Beacons, and to return consolidated knowledge using "equivalent concept cliques" to merge related Beacon results. The merging aggregation of concepts and statements relies on the data- and heuristic-driven identification of 'equivalent concept cliques' (https://github.com/STARInformatics/beacon-aggregator/blob/master/README.md).

## Programmatic access to the Knowledge Beacon API

A Python client library for accessing Knowledge Beacons can be used to access individual beacons. A sample Python script using this library is shown here below. Full details of the available beacon endpoint access methods is provided on the Github repository (https://github.com/NCATS-Tangerine/tkbeacon-python-client).

```python
#!/usr/bin/env python
from __future__ import print_function
from tkbeacon import build, KnowledgeSource
from tkbeacon.rest import ApiException
# create an instance of the BeaconApi class pointing at the remote
SMPDB beacon
b = build(KnowledgeSource.SMPDB)
try:
    concepts = b.get_concepts(categories = ['protein'], size = 10)
    for concept in concepts:
        print(concept.id, concept.name)
    print('All results are proteins:', all('protein' in concept.cate-
gories for concept in concepts))
except ApiException as e:
print("Exception when calling BeaconApi->get_concepts: %s\n" % e)
```

## Navigating Knowledge Beacons

The above Python client library harvests knowledge by navigating Knowledge Beacon through a chained series of REST API endpoint calls that return data as JSON formatted documents, annotated using the Biolink Model standards noted above.

Here we illustrate a modest example of a Beacon-based navigation of knowledge relating to a relatively rare human disease, Progressive Supranuclear Palsy. The example iterates with a sequence of queries over a pair of beacons, first returning lists of concepts with names matching keywords, then using the selected identifiers of identified concepts to retrieve related *subject-predicate-object* statement assertions, and finally, retrieving citation evidence supporting those assertions. The queries themselves are API calls returning JSON documents; however, here we summarize the sequence of these queries in simple tabular form (Tables 2 and 3).

The example first identifies that high levels of homocysteine appear to be associated with the disease, suggesting applying potential treatments against hyperhomocysteinemia, may mitigate some of the disease pathology. Potential treatments could include specific vitamin supplements (B12, B6), dietary interventions—such as betaine (from wheat, shellfish, spinach, and sugar beets), catechins (red wine, green, black and oolong teas, fruits like plum, apples, peach, strawberry and cherry, and beans and grains like broad bean, lentil and cocoa) including epicatechins (from dark chocolate) and garlic–along with regular exercise.

## Discussion

The Knowledge Beacon API (Fig 1) is a basic biomedical knowledge discovery interface that supporting a relatively simple high-level use case of user interaction with the biomedical knowledge space, and as such, lacks the fully expressive power of a general knowledge query language interface like SPARQL. However, the API contributes to FAIR data access in that for knowledge sources wrapped by it:

1. **Findability**: is enhanced by registries of API wrapped knowledge resources;

2. **Accessibility:** is facilitated by the use of the Swagger/OpenAPI encoding of the API, the public repository to the standard, and homogeneity of application of the API across knowledge sources. The API includes service endpoints to access metadata relating to the knowledge source and data encodings (see also Biolink Model below)

3. **Interoperability:** is promoted by the beacon data model alignment with the emerging Biolink Model of the Biomedical Translator Consortium, which provides for shared standards for concept identification (enforced use of CURIE namespaces) and for ontological

**Table 2.  Example of a basic sequence of beacon queries to discover knowledge about a human disease.**

| Step | Endpoint | Beacons[a] | Query | Result | | |
|------|----------|-----------|-------|--------|--------|--------|
| | | | | Type | Identifier | Description |
| 1 | /concepts | SEMMEDB, HMDB | keywords = "Progressive Supranuclear Palsy" | Concept | UMLS:C008868 | Progressive Supranuclear Palsy (disease or phenotypic feature) |
| 2 | /statements | HMDB | s = UMLS:C008868 | Statement | S1.1 | HMDB:HMDB000742 (Homocysteine)—related to -> Progressive Supranuclear Palsy |
| 3 | /statements/ details | HMDB | S1.1 | Evidence | PMID:20606437 | Levin J, *et al.* 2010: Elevated levels of. . . homocysteine in. . . progressive supranuclear palsy. Dement Geriatr Cogn Disord. 29(6):553–9 |
| 4 | /concepts | SEMMEDB | keywords = "hyperhomocysteinemia" | Concept | UMLS: C0598608 | Hyperhomocysteinemia (disease or phenotypic feature) |
| 5 | /statements | SEMMEDB | edge_label = treats & t = UMLS:C0598608 | Statement | S2.1.. S2.n | *subject*[1..n]—treats -> HMDB:HMDB000742 (Homocysteine) |
| 6 | /statements/ details | SEMMEDB | S2.x | Evidence | ? | ? |

[a] Beacons used in this example: SEMMEDDB: Semantic Medline Database, HMDB: Human Metabolome Database

**Table 3. The list of selected subject concepts from statement results from Table 2 example query step 5, given as inputs to example query step 6, to retrieve statement evidence of associated literature citations.**

| Identifier | Name | Description | Evidence (Excerpt of PMIDs) |
|---|---|---|---|
| UMLS: C0042845 | Vitamin B12 | Vitamin B12 functions as a cofactor for methionine synthase and L-methylmalonyl-CoA mutase. Methionine synthase metabolizes homocysteine. | 8416664; 8795472; 11266032 . . .many more citations |
| UMLS: C0087162 | Vitamin B6 | Vitamin B6 also plays a role in cognitive development through the biosynthesis of neurotransmitters and in maintaining normal levels of homocysteine | 8416664; 8416664; 11592439 . . .many more citations |
| UMLS: C0016410 | Folic Acid | Folic acid is the fully oxidized monoglutamate form of folate. | 8416664; 8676799; 11135086 . . .many more citations |
| UMLS: C0178638 | Folate | One of the most important folate-dependent reactions is the conversion of homocysteine to methionine in the synthesis of S-adenosyl-methionine. | 11348885; 10201405; 11902805 . . .many more citations |
| UMLS: C0051200 | Allicin | a bioactive component of garlic | 28810641 |
| UMLS: C0005304 | Betaine | Although a general name, historically applied to trimethylglycine (TMG) involved in methylation reactions and detoxification of homocysteine. | 15720203;18370637 |
| UMLS: C0007404 | Catechin | natural polyphenol and antioxidant, found in especially high levels in green tea, cocoa, and certain kinds of fruit (including wine) | 22899103 |
| UMLS: C0014485 | Epicatechin | stereoisomer of catechin, (–)-epicatechin is the most abundant flavanol found in dark chocolate | 22899103 |
| UMLS: C0015259 | Exercise | activity requiring physical effort, carried out to sustain or improve health and fitness! | 24923386 |

mapping of concept categories (e.g. "gene", "disease", etc.) and their associations (e.g. "gene to disease association") in the biomedical sciences (**https://biolink.github.io/biolink-model/**). The API also provides an endpoint for discovery of equivalent concept identifiers which aid in data integration across knowledge sources.

4. **Reusability:** is supported by the API linkage of knowledge statements to their "evidence" such as PubMed indexed literature. The metadata encodings used by the API, from the Biolink Model, are under a Creative Commons Zero v1.0 Universal license. However, the API is generally agnostic about the licensing of, and, does not propagate the deeper provenance information from, the underlying public knowledge sources it wraps.

A reference installation of Knowledge Beacons accessing various knowledge sources (Table 1) are currently running on a modest sized cloud server instance. Informal profiling of the performance of these reference Beacons indicates that their endpoints generally return data within a fraction of a second, with only occasional queries taking a second or two longer.

We also prototyped a "Knowledge Beacon Aggregator" to provide an enhanced asynchronous query/status/retrieval API as an integration layer for managing Beacon client access to, and merging of data from, a registered catalog of multiple Beacon implementations.

A basic question may be posed about the advantages or disadvantages of the Knowledge Beacon API relative to underlying native API's of knowledge sources such as the Monarch Biolink API (https://api.monarchinitiative.org/api/) [10] or CyREST (for NDex, https://github.com/cytoscape/cyREST) [15].

The core advantage of the Knowledge Beacon API versus native API's is the simplification of client software knowledge harvesting, merging and traversal of knowledge graphs (concept nodes and relationship edges) programmatically feasible, using an interface with a small number of endpoints, standardized output (JSON) data model, and systematic, extensible semantic encoding of concept categories and relationship predicates. One doesn't need to have *a priori* knowledge of the many heterogeneous, more numerous, or more complex native API formats across multiple target knowledge sources. Moreover, the Beacon API is semantically extensible

by community curation of the Biolink Model. Many native API's have hard coded semantics which can only semantically extended by the additional of new endpoints.

For example, the Monarch Biolink API already has well over 100 endpoints which, although well structured, require some prior knowledge of available path tags, which are a hardcoded set therefore semantic extension of the Biolink API for new data types would require the addition of new endpoints. Conversely, querying the Monarch Biolink API for finer grained biological concept types and relationships is likely to be more challenging than using the Biolink Model defined ontology mappings directly in queries of a Knowledge Beacon.

Conversely, it may be an interesting exercise (not yet attempted) to connect Knowledge Beacon API clients to existing graph data visualization tools and associated standards, such as Cytoscape [16], given that Beacons potentially export rich graph networks of concepts and relationships across disparate knowledge sources not already easily accessed by such tools (i.e. NDex).

In terms of relative disadvantages, although the wrapping of a given knowledge source as a Beacon only has to be "done once" by one community developer, to facilitate knowledge access by Beacon clients (as noted above) implemented by other members of the community, such wrapping of individual knowledge sources as Beacons remains a labor-intensive activity, despite the use of some off-the-shelf API generation standards, libraries and tools. This practical constraint currently limits the availability of publicly available Beacon implementations.

Aside from the Beacon developer needing to design and implement custom code against the particular native API, database or raw datasets of a given target knowledge source, the primary challenge is that the semantics of the knowledge source being wrapped must generally be heuristically translated from native encodings (if available) into the ontology mappings of the Biolink Model. This task is somewhat easier for knowledge sources which have a small number of easily resolved discrete data types (i.e. one or very few discrete Biolink Model concept categories of data) and namespaces with clear mapping onto those discrete data types. In contrast, more heterogenous "graph data" knowledge sources, for example, the NDex Bio biomedical network data archive (https://home.ndexbio.org/index/, wrapped by this project as the *ndex* Beacon), don't have such a clear concept category and relationship predicate tagging of much of the archived data. The development of useful but (so far) imprecise heuristics to tag such data on the fly is required to develop a useful Beacon. In other cases, such as biomedical knowledge resources whose data object namespace aggregates several types of concepts in a fuzzy manner with limited additional concept category tagging, it may be even more challenging to semantically tag data entries for beacon export.

A few common library and reference implementations are developed for Beacons; the Beacon platform would benefit from the further development of standardized tools to systematically assist such wrapping of native knowledge sources.

The availability of a shared API standard for knowledge integration doesn't, in and of itself, deal with all the challenges of FAIR data integration within the global community. Practical experience with knowledge harvesting using such API implementations has revealed performance issues relating to internet and service latency, bandwidth limitations. Knowledge warehousing in centralized knowledge graphs using ETL (Extract, Transform, Load) processes may sometimes result in a more tractable process for biomedical knowledge integration; however, such approaches are still faced with the task of merging equivalent concepts, including the elimination of duplicate concepts and the resolution of conflicting information, including weighting of assertions differing in levels of confidence. More unique to ETL warehousing approaches is the ongoing problem of keeping such resources up-to-date relative to their original knowledge sources. Note that ETL warehouses and API driven distributed knowledge harvesting approaches can be complementary, in that ETL data warehouses can also themselves

be accessed by the application of web service REST API's like the Knowledge Beacon API. In fact, some of the current Beacon implementations use this approach: a back end Biolink Model compliant Neo4j knowledge graph directly wrapped with the API.

Generally, API approaches to knowledge harvesting may work best with use cases involving smaller batches of knowledge retrieval based on a focused navigation of the knowledge space from larger open-ended data sources which would be refractory to import into centralized knowledge graphs.

## Acknowledgments

BMG and RMB collaborated on the predecessor "*Knowledge.Bio*" application embodying the knowledge harvesting activities implemented by the Knowledge Beacon API. CJM and BMG coined the name "Knowledge Beacon" to express the architectural vision of uniformly wrapped knowledge sources for distributed knowledge discovery and harvesting. RMB and his team elaborated the original software design of web service endpoints and initial code implementations embodying Beacon functionality, then guided further iterations of the API based on feedback from colleagues within the Biomedical Translator Consortium, with special mention to co-author VD who proposed insightful revisions to the API, based on his direct experience implementing a Beacon to wrap HMDB. Under overall supervision by RMB, the heavy lifting of iterative software development of Beacon implementations—including several beacons, client (including aggregator) and validation applications—was undertaken by LMH while he was a member of the STAR Informatics team, assisted by the valuable software programming contributions of several computing science cooperative education students: MG, IWS and KCHB.

The authors would like to sincerely thank Nomi Harris and Marcin Joachimiak of LBNL for their very helpful editorial feedback on, and suggested revisions to the draft manuscript.

The authors would also like to thank the various members of the Biomedical Data Translator Consortium who gave helpful user needs feedback and support of the Knowledge Beacon API during its development, in particular, Chris Bizon and Stephen Ramsay. We also acknowledge here Greg Stupp who, while employed at TSRI, implemented an earlier version of a Beacon wrapper for biomedical knowledge in Wikidata.

## Author Contributions

**Conceptualization:** Benjamin M. Good, Christopher J. Mungall, Richard M. Bruskiewich.

**Funding acquisition:** Christopher J. Mungall.

**Project administration:** Richard M. Bruskiewich.

**Software:** Lance M. Hannestad, Vlado Dančík, Meera Godden, Imelda W. Suen, Kenneth C. Huellas-Bruskiewicz, Richard M. Bruskiewich.

**Supervision:** Christopher J. Mungall, Richard M. Bruskiewich.

**Writing – original draft:** Richard M. Bruskiewich.

**Writing – review & editing:** Lance M. Hannestad, Vlado Dančík, Benjamin M. Good, Christopher J. Mungall, Richard M. Bruskiewich.

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
