## [Decision Letter · Decision Letter 0]

15 Jul 2020

PONE-D-20-09379

Knowledge Beacons: Web Service Workflow for FAIR Data Harvesting of Distributed Biomedical Knowledge

PLOS ONE

Dear Dr. Bruskiewich,

Thank you for submitting your manuscript to PLOS ONE and my sincere apologies for the delayed editorial decision. I regret to say that I have had a hard time securing reviewers, a common theme since the Covid-19 pandemic.

After careful consideration, we feel that it has merit but does not fully meet PLOS ONE’s publication criteria as it currently stands. Therefore, we invite you to submit a revised version of the manuscript that addresses the points raised during the review process.

In particular, both reviewers raise several issues (often with regards to presentation) that need to be carefully addressed in a revised version of the manuscript. I can highlight the need for more examples (a point mentioned by both reviewers) as well as some information on query performance: these are definitely of practical use to all researchers interested in adopting Knowledge Beacons. I trust you will find the reviewers' comments of value in improving your manuscript (see detailed reviews at the bottom of this email) and expect all their comments are adequately addressed in a revised version of your manuscript.

We look forward to receiving your revised manuscript.

Kind regards,

Vasilis J Promponas

Academic Editor

PLOS ONE

Journal Requirements:

When submitting your revision, we need you to address these additional requirements

"The Knowledge Beacon API was primarily developed as a subcontracted activity of the

Biomedical Data Translator Consortium, supported by funding provided by the National

Center for Advancing Translational Sciences (NCATS;https://ncats.nih.gov/), National

Institutes of Health, projects 1, 1OT2TR002584 and OT3TR002025 (VD at Broad) plus

1OT3TR002019 (all other authors).

The specification and associated software was developed largely independently of

NCATS management team technical oversight but reported to them and applied to the

research objectives of the project."

We note that one or more of the authors are employed by a commercial company: "STAR Informatics / Delphinai Corporation,"

Reviewers' comments:

Reviewer's Responses to Questions

**Comments to the Author**

1. Is the manuscript technically sound, and do the data support the conclusions?

Reviewer #1: Yes

Reviewer #2: Yes

2. Has the statistical analysis been performed appropriately and rigorously? 

Reviewer #1: N/A

Reviewer #2: N/A

3. Have the authors made all data underlying the findings in their manuscript fully available?

Reviewer #1: Yes

Reviewer #2: Yes

4. Is the manuscript presented in an intelligible fashion and written in standard English?

Reviewer #1: Yes

Reviewer #2: Yes

5. Review Comments to the Author

Reviewer #1: 1- Knowledge Beacons is an API (REST web service) so calling it a workflow is a little bit stretched. I would remove the term workflow from the title and abstract. It can be one of steps in a bigger workflow but the article is about the API, not the workflow.

2- FAIR part of the API is partially fulfilled. It is available (findable, accessible in public repo, versioned), uses community standards (OpenAPI...) but you don't use or enforce agreed controlled vocabularies or ontologies in order to increase interoperability (reusable in FAIR) with external resources (in a bigger framework). Since you are mentioning FAIR directly in the title, more effort should be put in describing why and how this API and its (meta)data are FAIR.

3- I would like to see more API example calls (on some website or described in git repository) and a discussion (advantages/disadvantages) of using this API instead of calling directly Monarch db APIs (BioLink API).

4- Discussion lines 221-232 should be moved away from the end of the discussion since it deals with design choices of other projects and misses the connection with the proposed API

5- End of discussion (lines 237-245) talks about two other projects (reasoner API ad explorer beacon) but again it does not fit with the rest of the work (presentation of the knowledge beacons API)

Reviewer #2: This paper describes the development of REST APIs to wrap 6 widely used knowledge sources (SemMedDB, Monarch DB, HMDb, Rhea DB, SMPDb and ndex) into knowledge beacons. This work serves as an initial first step towards the uniform integration of the resources mentioned herein in other software, which in turn will have a clear benefit towards the use of standardized vocabularies and relational structures in knowledge graphs - a common issue impeding the extensive usage for many of them in their current form - while serving the principles of FAIRness in data dissemination.

The manuscript does a very good work in describing the technical implementation of the beacons and also gives clear explanation of usage either with keywords or CURIE concepts. Results returned by a beacon follow the biolink model, which is a new and exciting effort towards standardazing and allowing the interoperability of knowledge graphs. The authors have also discussed any limitations of their method in the discussion section extensively.

Minor revisions

1. Even though the authors have not conducted an extensive assessment of the method's efficiency, it would be nice if they could provide 1-2 examples of query performance time for 2 different knowledge sources (since query times should be different for those). Perhaps a minimum and maximum recorded time on their setting would be a good indication (since indeed internet latency and bandwidth limitations can lead to big variations). It's undestandable that this implementation will not be as fast as some lower-level query interface, so providing some examples for query times will convince potential users that even though slower, its usage is not time-prohibitive and the benefits it provides are enough to overcome any issues of this type.

2. The authors mention in their paper that "conversion of Beacon statement results into RDF format is easily accomplished". Do the authors provide scripts for these conversions on their github page? If yes I think it should be referenced at that point, and if not they should point the reader to sourcecode that allows this conversion.

3. In the discussion section the authors refer to Linked Data Fragments, but I feel that this paragraph is lacking any connections to the rest of the manuscript and the project the way it is written. I would like to request from the authors to rewrite this paragraph so that the connection is clearer.

4. In the Results Section, subsection "Beacon implementations" numerous additional software are described in the first page, and the authors point only to the github pages of these. Are there any related publication where the authors could also point to for any of those? If yes, please add the references to those in that paragraph.

5. In the discussion section the authors mention the use of a "knowledge beacon aggregator". It would be nice if they could provide an example of how the aggregator can integrate the results from multiple beacons and what are the requirements for the results to be aggregated (or if this is described in detail in the github page of the tool, point to that description in that section).

I have two more suggestions rather than comments to the authors, which they can incorporate either in their manuscript or in their github page.

6. First, I would like to suggest to the authors to give a more concrete example of a biological application of their method, a use-case scenario where, for example, someone could use the API to extract knowledge and how that would look. Even though the technical implementation is very well described in the paper it would be nice if someone could see a more concrete example of the applicability of the knowledge beacon in real world scenarios. This can be easily incorporated into their already described sample workflow, where the two-step process is shown, but the usability of the results returned is not described in detail.

7. Following on the latter, and specifically for the ndex swagger, it would be nice to mention how this can be complementary to the core Cytoscape API, CyREST (Ono, Keiichiro, et al., 2015), since ndex goes hand in hand with Cytoscape.

6. PLOS authors have the option to publish the peer review history of their article (what does this mean?). If published, this will include your full peer review and any attached files.

Reviewer #1: No

Reviewer #2: **Yes: **Katerina C. Nastou

---

## [Author Response · Author response to Decision Letter 0]

14 Jan 2021

In attached document "Response to Reviewers".

---

## [Decision Letter · Decision Letter 1]

8 Feb 2021

Knowledge Beacons: Web Services for Data Harvesting of Distributed Biomedical Knowledge

PONE-D-20-09379R1

Dear Dr. Bruskiewich,

We’re pleased to inform you that your manuscript has been judged scientifically suitable for publication and will be formally accepted for publication once it meets all outstanding technical requirements.

Kind regards,

Vasilis J Promponas

Academic Editor

PLOS ONE

Additional Editor Comments (optional):

Reviewers' comments:

Reviewer's Responses to Questions

**Comments to the Author**

1. If the authors have adequately addressed your comments raised in a previous round of review and you feel that this manuscript is now acceptable for publication, you may indicate that here to bypass the “Comments to the Author” section, enter your conflict of interest statement in the “Confidential to Editor” section, and submit your "Accept" recommendation.

Reviewer #1: All comments have been addressed

Reviewer #2: All comments have been addressed

2. Is the manuscript technically sound, and do the data support the conclusions?

Reviewer #1: Yes

Reviewer #2: Yes

3. Has the statistical analysis been performed appropriately and rigorously? 

Reviewer #1: N/A

Reviewer #2: N/A

4. Have the authors made all data underlying the findings in their manuscript fully available?

Reviewer #1: Yes

Reviewer #2: Yes

5. Is the manuscript presented in an intelligible fashion and written in standard English?

Reviewer #1: Yes

Reviewer #2: Yes

6. Review Comments to the Author

Reviewer #1: I would suggest to remove the python client library sample script from main text and put it as supplementary information or as a link to a GitHub repository. The recovered space could be filled with some details about the library implementation and usability.

Reviewer #2: The authors have satisfactorily addressed all of my concerns and I am now happy to recommend this manuscript for acceptance.

7. PLOS authors have the option to publish the peer review history of their article (what does this mean?). If published, this will include your full peer review and any attached files.

Reviewer #1: No

Reviewer #2: No

---

## [Editor Report · Acceptance letter]

9 Mar 2021

PONE-D-20-09379R1 

Knowledge Beacons: Web Services for Data Harvesting of Distributed Biomedical Knowledge 

Dear Dr. Bruskiewich:

I'm pleased to inform you that your manuscript has been deemed suitable for publication in PLOS ONE. Congratulations! Your manuscript is now with our production department. 

Kind regards, 

on behalf of

Dr. Vasilis J Promponas 

Academic Editor

PLOS ONE